# COVID-19 Intensive Care—Evaluation of Public Information Sources and Current Standards of Care in German Intensive Care Units: A Cross Sectional Online Survey on Intensive Care Staff in Germany

**DOI:** 10.3390/healthcare10071315

**Published:** 2022-07-15

**Authors:** Anne Werner, Maria Popp, Falk Fichtner, Christopher Holzmann-Littig, Peter Kranke, Anke Steckelberg, Julia Lühnen, Lisa Marie Redlich, Steffen Dickel, Clemens Grimm, Onnen Moerer, Monika Nothacker, Christian Seeber

**Affiliations:** 1Department of Medical Psychology and Medical Sociology, University Medical Center Leipzig, 04103 Leipzig, Germany; 2Interdisciplinary Center for Health Sciences, Institute of Health and Nursing Science, Martin Luther University, Halle-Wittenberg, 06112 Halle, Germany; anke.steckelberg@medizin.uni-halle.de (A.S.); julia.luehnen@uk-halle.de (J.L.); lisa.redlich@student.uni-halle.de (L.M.R.); 3Faculty of Medicine, Department of Anaesthesiology, Intensive Care, Emergency and Pain Medicine, University of Wuerzburg, 97080 Wuerzburg, Germany; popp_m4@ukw.de (M.P.); kranke_p@ukw.de (P.K.); 4Clinic and Polyclinic for Anesthesiology and Intensive Care, Faculty of Medicine, University of Leipzig, 04103 Leipzig, Germany; falk.fichtner@medizin.uni-leipzig.de; 5TUM Medical Education Center, Faculty of Medicine, Technical University of Munich, 81675 Munich, Germany; christopher.holzmann-littig2@mri.tum.de; 6Department of Nephrology, Faculty of Medicine, University Hospital rechts der Isar, Technical University of Munich, 81675 Munich, Germany; 7Clinic for Anaesthesiology, University Medical Center of Göttingen, Georg-August-University Göttingen, 37085 Göttingen, Germany; steffen.dickel@med.uni-goettingen.de (S.D.); clemens.grimm@med.uni-goettingen.de (C.G.); omoerer@med.uni-goettingen.de (O.M.); 8Institute for Medical Knowledge Management c/o Philipps University Marburg, Association of the Scientific Medical Societies in Germany, 35043 Marburg, Germany; nothacker@awmf.org

**Keywords:** COVID-19, implementation, guideline usage, guideline adherence, intensive care, Germany, ICU staff

## Abstract

**Backround**: In February 2021, the first formal evidence and consensus-based (S3) guidelines for the inpatient treatment of patients with COVID-19 were published in Germany and have been updated twice during 2021. The aim of the present study is to re-evaluate the dissemination pathways and strategies for ICU staff (first evaluation in December 2020 when previous versions of consensus-based guidelines (S2k) were published) and question selected aspects of guideline adherence of standard care for patients with COVID-19 in the ICU. **Methods:** We conducted an anonymous online survey among German intensive care staff from 11 October 2021 to 11 November 2021. We distributed the survey via e-mail in intensive care facilities and requested redirection to additional intensive care staff (snowball sampling). **Results:** There was a difference between the professional groups in the number, selection and qualitative assessment of information sources about COVID-19. Standard operating procedures were most frequently used by all occupational groups and received a high quality rating. Physicians preferred sources for active information search (e.g., medical journals), while nurses predominantly used passive consumable sources (e.g., every-day media). Despite differences in usage behaviour, the sources were rated similarly in terms of the quality of the information on COVID-19. The trusted organizations have not changed over time. The use of guidelines was frequently stated and highly recommended. The majority of the participants reported guideline-compliant treatment. Nevertheless, there were certain variations in the use of medication as well as the criteria chosen for discontinuing non-invasive ventilation (NIV) compared to guideline recommendations. **Conclusions:** An adequate external source of information for nursing staff is lacking, the usual sources of physicians are only appropriate for the minority of nursing staff. The self-reported use of guidelines is high.

## 1. Introduction

By the end of 2021 the COVID-19 pandemic lasted for two years. Besides the emerging of effective vaccines and sufficient protective measures, specific therapies have been established. To accelerate the speed of the generated evidence to be applied at the bedside of COVID-19 patients, the evidence assessment network CEOsys (COVID Evidenz Ökosystem, www.covid-evidenz.de [1]) was funded by the German Ministry of Education and Research. This network consisting of 20 university hospitals with a diverse composition of experts, was centrally involved in developing and maintaining the living guideline recommendations for inpatient therapy of patients with COVID-19 in Germany [2]. This guideline was disseminated through various media channels (AWMF guideline registry—Association of the Scientific Medical Societies in Germany (Arbeitsgemeinschaft der Wissenschaftlichen Medizinischen Fachgesellschaften e. V), CEOsys website, MAGICapp, guideline pages and via participating scientific medical societies, webinar).

In the second half of 2021, a new wave of the delta Coronavirus variant reached Germany, leading to both increasing incidences and an increasing number of patients requiring intensive care treatment [3]. This wave threatened to overburden the German healthcare system, which had previously been able to keep incidences under control, limit the number of deaths and avoid a local collapse of the healthcare system.

As early as March 2020, at the beginning of the pandemic, a first guideline based on the informal consensus (S1) of a group of experts was published. In autumn 2020, the guidelines were raised (S2k) by one quality level until an evidence-based guideline with the highest quality level (S3) was finally published in February 2021 [2].

First evaluation of ICU treatment standards and dissemination preferences started in December 2020 [4]. Active distribution of evidence syntheses was preferred by all professions. To draw attention to medical information about COVID-19 participants wished a dissemination through public institutions, medical journals, scientific medical societies and newsletters. A re-evaluation of the dissemination channels, the CEOsys dissemination strategy and the actual care was conducted simultaneously with the publication of the 3rd version of the S3-guideline, as both the evidence base and the pandemic event had changed significantly. Therefore, we initiated an online survey similar to that at the end of 2020 to ensure a follow-up of the actual care and to re-evaluate the following dissemination strategy with a focus on intensive care units (ICUs) in Germany.

## 2. Materials and Methods

The study was designed in accordance with the Declaration of Helsinki and submitted to the ethics committee of the University of Würzburg (Ref: 2020-219/20), which did not require further review due to voluntary participation without intervention character. Reporting was done in accordance with the Checklist for Reporting Results of Internet e-Surveys (CHERRIES) [5] (Appendix A).

### 2.1. Recruitment and Participants

We conducted an anonymous online survey via SoSci-Survey [6] from 11 October 2021 to 11 November 2021 among German intensive care staff. The survey was distributed via e-mail in intensive care facilities of Würzburg, Göttingen, Leipzig and also in the surrounding hospitals with intensive care units. At the same time, all CEOsys members (*n* = 124) were asked to forward the survey link to their local ICUs across Germany. The link of this open-design survey was to be shared with the intensive care staff of each facility (snowball sampling [7]). A reminder was sent on 27 October 2021. Initially over 700 people were contacted. To ensure anonymity, no cookies were used to identify individual users, no IP controls or log file analysis were performed and participants were not asked to register or verify themselves.A previous survey had been published one year earlier; information on participants, survey period and procedure can be found there [4].

### 2.2. Questionnaire

We used a self-constructed questionnaire that was divided into three sections and consisted of 12 screen pages: demographics (seven items), quality, barriers and trust in sources of information (maximum 13 items, addaptive structured) and guideline adherence to treatment standards (three main items and 12 additional items). The questionnaire used and adapted seven items from a survey conducted in December 2020 (published in Seeber [4] and Dickel [8]). The order of the items was not randomised. Due to the assumed shortage of time of the respondents during the pandemic situation we decided to use a short questionare with mainly categorical variables and voluntary free text options.

The section “Quality, barriers and trust in information sources” contained questions about use and quality assessment of eight given sources (Appendix A). Participants had the option to specify the selected sources (multiple answers possible, no restriction). Furthermore, they were able to assess the quality of all eight given sources, using a numerical rating scale from 0 to 10, with 10 representing the highest possible quality. Additionally, they were asked whether they use the clinical practice guidelines, the CEOsys website (www.covid-evidenz.de [1]) and MAGICapp (a digital authoring and publication platform, https://magicevidence.org/ [9]). In a next step, the participants were asked about recommendations, barriers and the applicability of these three sources. They also could choose one to three out of eight healthcare organizations, which they trust most in relation to the ongoing pandemic and the dissemination of knowledge. Three questions were asked about compliance with the standards of care: discontinuation criteria for non-invasive ventilation (NIV), dosing of thromboembolic prophylaxis and medications currently used for COVID-19 therapy. In addition, there were 13 items to specify the answers.

The exact wording, ranking, and response instructions for the question about NIV discontinuation criteria were taken from the 2020 online survey by Seeber et al. [4]. So far, the results on this question have not been published. The second unpublished question asks about agreement with the statement “Severely ill COVID-19 patients should be treated with therapeutic anticoagulation/blood thinning” using a 4-point Likert scale from “I do not agree” to “I agree”. For the analysis, respondents’ answers were dichotomized into “disagree” and “agree”. We pre-tested the questionnaire for plausibility and comprehensibility with four representatives (two physicians, two nurses) of the addressed professional groups and modified items based on their comments. The final survey can be found in the Appendix A.

Just before the start of the survey, the current evidence based “S2” clinical practice guideline “Recommendations for inpatient therapy of patients with COVID-19” was published in Germany on 5 October 2021 [8]. To disseminate the new version of the guideline a press release was issued in the German Medical Association’s official journal (Deutsches Ärzteblatt) and in internal distribution lists of the 16 participating scientific medical societies.

### 2.3. Data Analysis

We used descriptive statistics (percentages, median (MD), interquartile range (IQR)) to report the results. By using the median as a measure of central tendency, the sensitivity to outliers was reduced [10]. As far as possible, free texts were categorized, summarized and expressed in absolute frequencies of mentions. Due to the sampling method, the small number of participants, the non-surveyed demographics and the suspected reduced representativeness of the data, an inferential statistical analysis was not carried out [11,12]. All analyses were performed with SPSS 27. To verify additional assumptions to free open access medical education (FOAM) we used Spearman correlation (age and quality rating) and Mann-Whitney-U-Test (quality assessment between users and non-users).

## 3. Results

The online questionnaire was started by 156 Persons and completed by 104 (completion rate 66%). We also included incomplete questionnaires in our analyses. Completing the questionnaire took an average of 5.5 min.

### 3.1. Participants

The first substantive question was completed by 130 participants. 26 medical residents, 50 specialists, 20 nurses, 31 intensive care nurses and three members of other health professions participated. The last group was excluded for all calculations due the small group size. If residents and specialists are meant, the term “physicians” is used; if nurses and critical care nurses are meant, the term “nursing staff” is used. For age characteristics see Appendix A.

127 participants were included in statistical calculations. At the time of the survey, the majority worked in maximum care hospitals (*n* = 110, 86.6%). Only 17 participants (13.3%) worked in primary care hospitals. Appendix A shows the age distribution of the participants. Overall *n* = 43 (33.9%) held a position as head physician/head nurse setting up treatment plans. 24 participants had worked in the COVID-19 Evidence Ecosystem project (CEOsys).

The number of participants in the 2020 survey was larger overall but the distribution was very similar [4].

### 3.2. Use, Quality, Barriers and Trust in Information Sources

Participants mostly preferred Standard Operating Procedures (SOPs)/procedural instructions (*n* = 99, 78%), medical journals (*n* = 75, 59.1%) and websites of scientific medical societies (*n* = 64, 50.4%). Everyday-media (*n* = 47, 37.0%), medical information portals (*n* = 40, 31.5%), newsletters/e-mails (*n* = 27, 21.3%), free open access medical education (FOAM) (*n* = 21, 16.5%) and social media (*n* = 16, 12.6%) were less favoured. Especially regarding the websites of scientific medical societies, FOAM, medical journals and medical information portals, the responses varied between professional groups (Figure 1). Medical residents and intensive care professionals used more sources (*MD* = 4) than nurses and intensive care nurses (*MD* = 2).

Table 1 shows the free text specifications of the users of medical journals, websites of scientific medical societies, everyday-media, social-media, FOAM and medical information portals.

Less than one in four nursing staff used medical journals (*n* = 11, 21.6%), while more than three in four physicians (*n* = 64, 84.2%) used this information source (Figure 1). Seven from these eleven nursing staff reported using only one magazine and only German-language journals (only one used an international journal, 50% of these German journals had a nursing focus), while most physicians used more than one journal (MD = 4) and mainly international journals (82 from 145 nominations).

The quality rating was similar to the frequency of use, except for FOAM (Table 2). For everyday-media, there is a difference in the quality rating between the professional groups of physicians (MD = 2, IQR = 1–4) and nursing staff (MD = 4, IQR = 1–6).The websites of the scientific medical societies, medical journals and in-house SOPs/procedural instructions received the best quality ratings in all professions (each MD = 8). Overall, nurses used the COVID-19 information sources listed in the survey less frequently than physicians, but their ratings were comparable.

Table 3 shows group differences in quality rating for users and non-users of this source. The effect size varies from strong to weak. In particular, for less used sources, there is a strong effect size for quality rating. There is no significant difference in the age category of those who did not know FOAM well enough to evaluate it and those who did (U = 1507.0, Z = −1.649, *p* = 0.09) (Appendix A). But in general there is a significant correlation between age category and qualityrating of FOAM (ρ = −0.372, *p* = 0.002).

Figure 2 shows that trust in different organisations providing information on COVID-19 therapy varies according to the professional group.

### 3.3. Evaluation of the Guideline, MagicApp and CEOsys Website

All professional groups reported a high usage of guidelines (total *n* = 99, 86.8%): medical residents 83.3% (*n* = 20), specialists 95.7% (*n* = 45), nurses 80% (*n* = 15) and intensive care nurses 78.6% (*n* = 28). The majority of participants rated the applicability of the guidelines as rather good (*n* = 76, 76.8%) or very good (*n* = 13, 13.1%). No one rated rather poor or very poor. Ten participants (7.9%) selected neither good, nor bad applicability and selected “undecided”. This small group criticised the need for faster updates (*n* = 2), insufficient integration of various recommendations into the overall concept (*n* = 1), lack of patient compliance (*n* = 1) or a poor a fit between guideline and outcome (*n* = 1) and weak evidence (*n* = 1).

Only 11.1% (*n* = 12) of the participants used MAGICapp, 10 out of these 12 participants were CEOsys employees. All users of MAGICapp recommended the usage and rated the applicability as rather good (58.3%), very good (33.1%) or undecided (8.3%). The reason given for the “undecided” rating was: “must be familiar with the structure”. The CEOsys website was also rarely used (*n* = 17, 15.5%). 16 out of 17 users recommended the Website. The reported barriers to using guidelines, MagicApp and the CEOsys website were similar. Mainly, there was a lack of time for using these sources (CEOsys 46.7%, Guidelines 53.3%, MAGICapp 22.6%), lack of experience (CEOsys 26.7%, Guidelines 20%, MAGICapp 49.5%) and a lack of knowledge about the accessor as the website was unknown (CEOsys 65.6%, Guidelines 33.3%, MAGICapp 46.2%).

### 3.4. Compliance in Treatment Standards

Nearly 70% of the participants (*n* = 88, 69.3%) reported an established standard of care for patients with COVID-19. Approximately half of the participants reported standard administration of prophylactic dosing of thromboembolic prophylaxis (*n* = 52, 51%), about a quarter reported semitherapeutic dosing (*n* = 24, 23.5%) and therapeutic dosing (*n* = 26, 25.5%) in patients with COVID-19 pneumonia in the ICU (Appendix A). This is predominantly controlled with laboratory coagulation monitoring (79.2% for semitherapeutic dosing, 96% for therapeutic dosing). In 2020 most participants (*n* = 198, 82.8%) agreed to therapeutic anticoagulation of patients with COVID-19 pneumonia. The agreement among nurses (*n* = 70, 80.5%) was similarly as high as that among physicians (*n* = 128, 84.2%).

The most frequently cited discontinuation criterion for NIV is impaired consciousness (*n* = 82, 78.8%), followed by the Horovitz Index (*n* = 77, 74.0%), clinical assessment of respiratory work (*n* = 73, 70.2%), CO_2_-elimination disorder (*n* = 58, 55.8%) and respiratory rate (*n* = 41, 39.4%). Less than 20% of the participants named Work-of-breathing-Index (*n* = 19, 18.3%) and Rapid-shallow-breathing-Index (*n* = 12, 11.5%). While the choice “disturbance of consciousness” was similar across all professions, they differed in their choice of Horovitz-Index and clinical assessment of respiratory work (Table 4). Intensive care specialists in particular selected these two items less frequently than medical residents. In percentage terms, Horovitz-Index and clinical assessment of respiratory work was selected more frequently than in the 2020 pre-survey. Overall the distributions in 2020 and 2021 are similar. In both surveys, “I don’t know” was choosen by 4.8–6.5% of the participants.

Most commonly used medication for treatment of COVID-19, the rationale for therapeutic consideration and the recommendation in the different versions of the guideline are shown in Table 5. The use reported by participants was based on the guideline for the most common five medications, except in case of vitamin D. In this case supervisor instructions in 41.7% and guideline use in 33.3% were the rationale of the intervieews for therapy. Although participants referred to the guideline on using vitamin D and remdesivir, there was no supporting recommendation.

## 4. Discussion

Our study revealed differences in the use of various COVID-19 information sources between different subgroups of ICU professionals (medical residents, specialists, nurses, intensive care nurses) regarding treatment for COVID-19 patients. The most frequently cited sources of information were SOP’s, medical journals and websites of scientific medical societies. The quality of the sources was rated similarly across all professional groups. The most trusted organisations for disseminating new medical information were the German public health institute (Robert Koch Institute) and scientific medical societies. The Cochrane Collaboration was only known by physicians. The use of the evidence-based guideline was ranked high by the interviewees and applicability was rated as good. The dissemination of the MAGICapp and CEOsys websites needs to be improved. The actual care of COVID-19 patients was reported to be largely in accordance with the recommendations of the existing guideline regarding the use of NIV, anticoagulation, and specific medications. As this survey was a follow-up to our recently published survey [4], we found some similarities and examined some detailed questions. Due to the low participation, the non-collected demographic data and the different number of answer options, the results could not be directly compared.

Even though nursing staff rate the quality of information sources very similarly to physicians, they use most of these sources less. Only SOP’s are used extensively by both professional groups. The frequent use of everyday media by nursing staff compared to that of professial medical sources is consistent with the findings of Cheese [12]. Nursing staff rated the quality of newsletter/email and everyday-media better than physicians, but used emails less frequently, just as physicians did. Perhaps it is more difficult to integrate these into the daily work routine. All professional groups rate the quality of social media as being low and rarely use it. This is consistent with the findings of Tunnecliff [13] and Falcone [14].

While physicians are reached through more and different channels (SOPs, medical information portals, websites of the scientific medical societies, medical journals), nursing staff are reached less and mainly through SOPs and everyday media. The fact that nurses use fewer and different strategies than physicians is consistent with the research by Jordan et al. [15]. This could be based in the information dissemination strategy. Users have to actively search for specific information on websites, information portals and medical journals (“pull strategy”). Everyday media can be consumed more passively (“push-strategy”). Seeber et al. reported that 68.6% of nursing staff (*n* = 70) and 74.6% of physicians (*n* = 130) preferred a “push strategy”. Also in this study, physicians reported using professional journals (pull strategy) more frequently than nursing staff, while nursing staff used social media and everday media (push strategie) more frequently than physicians. A push strategy may not be substantial enough. Studies have shown that passive provision of information can increase awareness, but does not change physicians’ behaviour [16,17,18].

There is a lack of high quality public information sources that address the information needs and opportunities of nursing staff. More qualitative baseline research (e.g., focus group interviews) may be needed to identify needs, barriers and opportunities in nurses’ information behaviour. The systematic review by Mostofian et al. [16] shows that there are only a few reviews and surveys that address the effectiveness of providing information to nursing staff.

The quality assessment and use of FOAM was striking. Only 56 participants knew this source well enough to rate the quality. Both professional groups rated the quality to be high but rarely used it. Contrary to our assumption, there was no significant difference in the age of those who did not know FOAM well enough to evaluate it and those who did. In general, there is a significant correlation between age category and quality rating of FOAM. FOAM tends to be an ambivalent source of informationbecause it is not subject to a review process. In addition, quality differences between individual FOAM sources can not be ruled out, which may have led to such a different evaluation. On the other hand, the content is often presented clearly and plainly [19,20] and might be easily accepted by readers especially at the beginning of their medical career.

There are significant differences in quality assessment between those who use an information source and those who do not not. With the exception of FOAM and medical journals, all of these sources are more passively consumable. It is not surprising that people use information that they consider to be of high quality or that they do not use COVID-19 information sources they consider to be of low quality. Similar results were already found by Falcone et al. in their study of the information needs of the italian population [14]. However, it is remarkable that this effect is particularly stronger for less frequently used sources like FOAM and social media.

Most participants reported using the evidence based guideline-recommendations for inpatient therapy of patients with COVID-19. During our survey period, the guideline was updated [10]. Most participants estimateds the applicability as good or very good. Thus, the guideline has a high self-reported usage rate, yet for physicians this rate is 91.5%, but for nursing staff only 79.1%. Barriers of guideline use were lack of time in the daily work routine and difficult access. Lack of time is also afrequently mentioned barrier of guideline implementation in the literature [4,21,22,23,24,25].

For all five of the most commonly used treatments, guidelines are the primary basis for decision-making except for vitamin D. Vitamin D was used by 24 participants (14 physicians, 10 nursing staff) and is thus the third most frequently used drug, although there has been a negative recommendation in every guideline update since February 2021. In contrast, recommended corticosteroid therapy was well established in this group (23 of 24, 95.8%). At this point, we did not assign the participants to individual ICUs and therefore cannot say whether these 24 do not all work in the same ICU. For drugs with changing or unclear data at the time of collection [2,10,11], such as remdesivir, IL6 receptor blockers, and JAK inhibitors, we see noncompliant use, although the guidelines are cited as the basis for decision-making. The narrow time frame for indicating JAK inhibitors and IL-6 receptor blockers may be one reason. Reasons for using remdesivir outside the scope of the guideline should be further investigated.

In a first CEOsys survey in 2020, before the first guideline was published, the results for discontinuation critera for NIV were very similar. The guideline only explicitly mentions Horovitz Index and respiratory rate as discontinuation criteria [2,10,11], so a more frequent mention of these two criteria would be expected. While the Horovitz Index is one of the most frequently selected criteria and the relative frequency of mentions increased between the two surveys, respiratory rate was mentioned with similar frequency compared with 2020 (about 40%). Especially nursing staff who regularly apply this therapy, choose this item very seldomly. However, nurses on intensive care units need to manage NIV therapy temporarily on their own. For this decision making they need a solid knowledge base. Thus, utilisation has not been considered in this professional group and should be emphasised more in the local recommendations for standard care. In a CEOsys survey of leading ICU physicians in 2020, participants selected all answers very frequently [8]. So here we may see a difference in perceived standards of care and lived reality.

Most of the differences could be seen in the answers of the nursing staff. Since these occupational groups represent the smaller groups of participants (in 2020 and 2021), the conclusions here are limited.

At the onset of the pandemic in early 2020, there were data indicating an increased incidence of thrombosis and pulmonary artery embolism [26]. Therapeutic anticoagulaton appeared to reduce these specific complication of COVID-19. In the fourth quarter of 2020, 85.8% (*n* = 205) of participants agreed to therapeutic anticoagulation of patients with COVID-19-pneumonia. One year later there was only one in four who reported this therapeutic use. This shows the rapid change in behaviour. Whether or not this change was achieved by the S3 guideline could not be clearly confirmed.

Thus, despite self-reported high use and a predominantly good to very good applicability of the guideline as perceived, the participants reported deviations from guideline recommendations in the application questions. The therapeutic use of non-recommended medications like vitamin D and remdesivir, the comparatively rare use of recommended IL-6 receptor blockers and JAK inhibitors, as well as the relatively low frequency with which the respiratory rate was mentioned as a criterion for discontinuing NIV are striking.

Based on a model of general mechanisms of action for guidelines, Cabana et al. [22] classified barriers into three categories: knowledge, attitudes and behaviour [22]. The majority of participants stated that they used and recommended the guidelines. So barriers are not due to a lack of agreement or a lack of awareness (knowledge). But there may be barriers in attitude: inertia in previous practice. Thus, local standards, structures, habits, and routines may be stronger than the recommendations of the guidelines. [22,27]. It is also conceivable that the answer to the question about the utilization of the guideline is strongly determined by social desirability [28]. Further more standardized investigations are needed [29].

So were the German standards of care for intensive care units being met? Ina Kopp defines guideline compliance as a measure of the conformity of an actor’s knowledge, thought or action with the recommendations and quality objectives given in a guideline [30]. Thus, according to this definition, there was high guideline compliance. While there were some differences in reported behaviour, there was a high level of agreement on the use and applicability of the guidelines. We did not measure any patient-related influencing factors (e.g., comorbidities, expectations, cultural background) that may affect the behaviour of physicians and nursing staff [28,30]. In addition, structural and outcome quality endpoints should be investigated and included to assess guideline quality and barriers in usage [31].

There were some limitations to our study. (1) We used a small, nonrepresentative, non-randomised sample. There was a very low estimated response rate to our invitations although email recruitment is actually considered a way to minimize the nonresponse error [32]. The low estimated response rate could be due to the increasing workload during the survey period and the increasing number of COVID-19 patients requiring intensive care, but also to a decreasing willingness to participate in COVID-19 studies or surveys in general. The high number of participants from maximum care hospitals is due to the recruitment process. Recruitment was not evenly structured across Germany, instead it was based only on personal contacts of the (co-)authors. (2) We do not have data on how our sample is distributed locally. (3) The separation between everyday media and social media is problematic. For most people, social media is part of everyday life. Reliable everyday media such as the public broadcaster ARD can also be received via social media channels such as Instagram or Twitter. (4) The separation between FOAM and social media is problematic as Twitter can be used to disseminate information you can find on FOAM [33]. (5) We only asked about general public sources of information, not for specific German sources such as the AWMF guideline register. If there had been questions on specific sources, participants would have been able to evaluate more precisely what they use and how they assess quality.(6) There were three questions to measure adherence in treatment standards. These three items were only self-reported behaviours, without knowledge of the circumstances or the specific use case that the participants may have had in mind when answering them. An objective study of the behaviour of medical staff (e.g., medical parameters and case studies) would be desirable.

## 5. Conclusions

The local standard of care in form of a fixed procedure was ranked as the most important tool by the majority of the intensive care teams. Especiallly for most nursing staff, who had less professional sources of information, but who needed to apply and monitor therapeutic strategies at the bedside the local written standard should have been the key of knowledge transfer. An adequate external source of information for nursing staff is lacking, the usual sources of physicians are only appropriate for the minority of nursing staff. Active measures of each facility to widespread the correct use of the local standards may be mostly accepted. The dissemination of information must therefore also reach those who are responsible for the preparation of the SOPs. Trustworthiness in organizations did not vary much over the time. Reported use of the evidence-based guideline “Recommendations for inpatient therapy of patients with COVID-19” took place regularly, was rated good but showed some deviation in the implementation. In further investigations, the reasons for the deviation should be recorded. Therefore applying the evidence based measures of the guideline to the local standard should be a focus of the implementation strategy.

## Figures and Tables

**Figure 1 healthcare-10-01315-f001:**
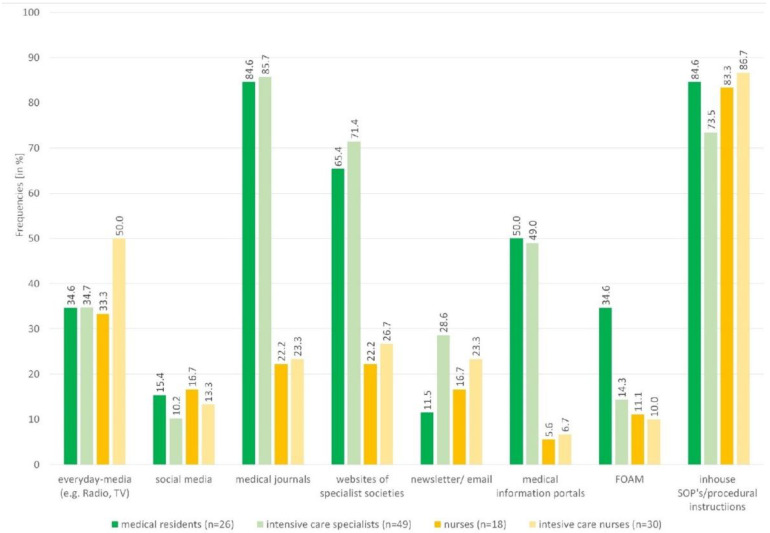
Use of information source according to profession. *SOP* standard operating procedure, *FOAM* free open access medical education.

**Figure 2 healthcare-10-01315-f002:**
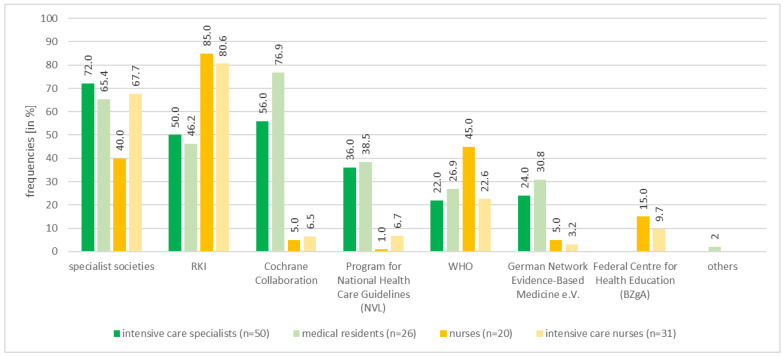
Trust in organizations according to professions 2021. *RKI* Robert Koch Institute, *WHO* World health organization.

**Table 1 healthcare-10-01315-t001:** Media sources used by participants, summed up free text answers.

Medical Journals					
International Journals	Physicians	Nursing Staff		Physicians	Nursing Staff
	n	n	German Journals	n	n
*NEJM*	21	1	*Deutsches Ärzteblatt*	16	
*The Lancet*	17		*AINS*	11	
*JAMA*	15		*Anästhesiologie & Intensivmedizin*	10	
*Critical Care*	7		*Der Anästhesist*	6	1
*Intensive Care Medicine*	7		*Intensiv*		2
*The BMJ*	3		*Die Schwester der Pfleger*		2
*Critical Care Medicine*	3		*Notfall + Rettungsmedizin*		1
*European Journal of Anaesthesiology*	2		*PflegenIntensiv*		1
*Anesthesiology*	2		*Intensivpflege*		1
*Nature*	2		*Intensiv news*	1	
*Anesthesia & Analgesia*	1				
*AJRCCM*	1				
*Journal of Intensiv Care*	1				
*NEJM* The New England Journal of Medicine, *JAMA* Journal of the American Medical Association, *AJRCCM* American Journal of Respiratory and Critical Care Medicine, Anesthesiology, *AINS* Anästhesiologie, Intensivmedizin, Notfallmedizin, Schmerztherapie					
Websites of scientific medical societies ^a^					n
*DIVI*					27
*DGAI*					20
*AWMF*					13
*DIVI* German Interdisciplinary Association for Intensive and Emergency Medicine (Deutsche Interdisziplinäre Vereinigung für Intensiv- und Notfallmedizin), *DGAI* German Society of Anaesthesiology and Intensive Care Medicine (Deutsche Gesellschaft für Anästhesiologie und Intensivmedizin), *AWMF* Association of the Scientific Medical Societies in Germany (Arbeitsgemeinschaft der Wissenschaftlichen Medizinischen Fachgesellschaften e. V.),					
everyday-media ^b^					
Public broadcasters (e.g., *ARD, ZDF, BBC*)					14
Daily/weekly newspapers (incl. online editions) (e.g., *Der Spiegel, New York Times, FAZ, Die Zeit*, regional daily newspaper, *Süddeutsche Zeitung*)					14
TV in general					12
Social media ^c^					
*Twitter*					6
*Instagram*					5
*Facebook*					2
FOAM ^d^					
*EmCrit.org*					6
*nerdfallmedizin.blog*					6
*pin-up-docs.de*					4
Medical information portals ^e^					
* www.uptodate.com *					14
*AWMF*					7
*AMBOSS*					5

^a^ mentioned 6 times or less: Professional association of German anaesthesiologists (Berufsverband Deutscher Anästhesisten), RobertKochInstitute, German Society for Internal Intensive Care and Emergency Medicine e. V. (Deutsche Gesellschaft für Internistische Intensivmedizin und Notfallmedizin e. V.), European Society of Intensive Care Medicine, European Society of Anaesthesiology, European Society of Anaesthesiology and Intensive Care, Association of Anaesthesia Associates, Infectious Diseases Society of America, American Society of Anesthesiologists, ^b^ mentioned 6 times or less: radio, internet, podcasts, Reuters news agency, ^c^ each mentioned once: Jodel, Linkedin, ReasearchGate, Telegram, ^d^ mentioned 2 times or less: dasFOAM.org, young urban anesthesiologists, fasttrack-notfall.com, Thieme CNE, uptodate.com, rebellem.com., ^e^ mentioned 2 times or less: Pubmed, springermedizin.de, Thieme, CEOsys, MAGICapp, DocCheck, Medscape.

**Table 2 healthcare-10-01315-t002:** Median, interquartil range and percentage of usage of information sources over all participants.

	Physicians (*n* = 76)			Nursing Staff (*n* = 51)			Total (*n* = 127)		
	Use	Quality *		Use	Quality *		Use	Quality *	
	%	MD	IQR	%	MD	IQR	%	MD	IQR
Social media	12.0	1	0–3	16.6	2	0–3	13.0	1	0–3
Everyday media (e.g., Radio, TV)	34.7	2	1–4	43.8	4	1–6	38.2	2.5	1–5
Newsletter/email	22.7	5	4–7	20.8	7	6–8	22.0	6	4–8
FOAM	21.3	7	5–7.5	10.4	7	5–8	17.1	7	5–8
Medical information portals	49.3	7	6–9	6.3	8	6–8.75	32.5	7	6–9
Websites of the scientific medical societies	69.3	8	7–9	25.0	8	6–9	52.0	8	7–9
Medical journals	85.3	8	8–9	22.9	8	7–10	61.0	8	7–9
Inhouse SOP’s/procedural instructions	77.3	8	7–9	85.4	8	7–9	80.5	8	7–9

*** MD and IQR for those who know the source well enough to assess the quality, various *n* (*n*= 119 for SOP’s—*n* = 87 for FOAM), Range: social media 0–9, everyday-media/newsletter/FOAM 0–10, medical journals/websites/medical information portals/ SOP’s 1–10.

**Table 3 healthcare-10-01315-t003:** Group differences in quality assessment of information sources between users and non-users (Mann-Whitney-U-Test).

	*p*	r
Social media (*n* = 98)	<0.001 *	−0.49
Everyday media (e.g., Radio, TV) (*n* = 116)	<0.001 *	−0.43
Newsletter/email (*n* = 94)	<0.001 *	−0.38
FOAM (*n* = 56)	<0.001 *	−0.45
Medical information portals (*n* = 87)	0.075	−0.19
Websites of the scientific medical societies (*n* = 100)	0.012	−0.25
Medical journals (*n* = 106)	0.006*	−0.27
Inhouse SOP’s/procedural instructions (*n* = 119)	0.206	−0.12

* significant *p* < 0.05.

**Table 4 healthcare-10-01315-t004:** Percentage of use of different discontinuation criteria for NIV for different occupational groups.

	Disturbance of Conscious-ness		Respiratory Rate		Clinical Assessment of Respiratory Work		Rapid-Shallow-Breathing-Index		CO_2_-Elimination Disorder		Horovitz-Index/Oxygenationindex		Work-of-Breathing-Index		I Don’t Know	
	2021	2020	2021	2020	2021	2020	2021	2020	2021	2020	2021	2020	2021	2020	2021	2020
	%	%	%	%	%	%	%	%	%	%	%	%	%	%	%	%
Medical residents	78.3	72.5	47.8	40.6	87.0	62.3	4.3	10.1	56.5	53.6	91.3	66.7	13.0	11.6	4.3	8.7
Intensive care specialists	81.0	82.1	61.9	53.0	61.9	65.8	16.7	13.7	66.7	62.4	69.0	70.1	21.4	12.0	4.8	2.6
Nurses *	83.3	61.5	8.3	28,2	50.0	46.2	16.7	5.1	25.0	51.3	75.0	48.7	16.7	25.6	8.3	12.8
Intensive care nurses	74.1	58.2	11.1	44.3	77.8	51.9	7.4	20.3	51.9	51.9	66.7	55.7	18.5	21.5	3.7	7.6
Total	78.8	71.1	39.4	44.7	70.2	58.9	11.5	13.5	55.8	56.3	74.0	62.8	18.3	16.1	4.8	6.5
Leading ICU physicians 2020 [7]		87.9		81.8		85.5		27.9		77.6		82.4		13.9		

2021 (*n* = 104): medical residents (*n* = 23), specialists (*n* = 42), nurses (*n* = 12), intensive care nurses (*n* = 27) 2020 (*n* = 304): medical residents (*n* = 69), specialists (*n* = 117), nurses (*n* = 39), intensive care nurses (*n* = 79) leading physicians in ICU 2020 *n* =165 [7] * in 2021 there were 20 less participants—conclusions are limited.

**Table 5 healthcare-10-01315-t005:** Recomendations from 3 guideline versions in 2021, frequencies of usage and rationale for decision-making.

	Recommended in Guideline… [1,8,9]			Survey 2021 ^a^	Rationale for Decision Making in % (for n Selections)				
	Feb 2021	May 2021	Oct 2021	n (%)	Evidence Based S3 Guideline	Own Literature Research	Determined by Supervisors	Good Experience with This Medication	I Do Not Know Why
Corticosteroids (e.g., dexa-methasone)	*p*	*p*	*p*	97 (93.3)	63.9	3.1	23.7	2.2	7.2
IL-6 receptor blockers (e.g., tocilizumab, sarilumab)	n	*p* ***	*p*	29 (27.9)	70	10	13. 3	-	6.7
Vitamin D	n	n	n	24 (23.4)	33.3	12.5	41.7	-	12.5
Specific antibodies	n *	n *	n **	23 (22.1)	54.5	18.2	18. 2	-	9.1
Remdesivir	-	nn	nn	17 (16.3)	47.1	17.6	17.6	5.9	11.8
JAK inhibitors (e.g., baricitinib)	-	-	*p*	9 (8.7)	66.7	11.6	22.2	-	-
Convalescent plasma	n	n	n	3 (2.9)	33.3	0	66.7	-	-
Ivermectin	n	n	n	2 (1.9)	50	50	-	-	-
Hydroxy-chloroquine	n	n	-	1 (1.0)	0	100	-	-	-
Lopinavir/ritonavir	n	n	-	2 (1.9)	50	50	-	-	-
others ^b^				6 (5.8)					

*n* = 104. n = negative recommendation, nn= neither positive nor negative recommendation, *p* = positive recommendation, * bamlanivimab, ** casirivimab & imdevimab, *** for subgroups ^a^ 11 October 2021 to 11 November 2021, ^b^ in free text mentioned: budesonid, vitamin C, vitamin B1 *IL-6 receptor blockers* Interleukin 6 receptor blockers, *JAK inhibitors* Janus kinase (JAK) inhibitors.

## Data Availability

The dataset used and analyzed during the current study is available from the corresponding author on reasonable request.

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
