# Peer review of "COVID-19 Intensive Care—Evaluation of Public Information Sources and Current Standards of Care in German Intensive Care Units: A Cross Sectional Online Survey on Intensive Care Staff in Germany"

_healthcare, 2022, doi:10.3390/healthcare10071315_

Round 1

Reviewer 1 Report

The topic is of interest, although there are several methodological concerns (e.g. related to the recruitment procedure) and the fact that results are only presented descriptively. The data do not offer any further chances for in-depth analyses. Nevertheless, the data provide some important insights, even though many different aspects have been assessed.

I recommend the following changes:
-       Abstract: The background section in the abstract has no headline.
-       Abstract: It might be irritating when you claim that the guideline has been published in February 2021, but already evaluated in December 2020.
- Abstract: The methods section is too short. Further information about recruitment and content of questionnaire is needed.
- Introduction: There are references missing in the first two paragraphs.
-   Introduction: The statements related to Declaration of Helsinki, ethics, and checklists should be part of the methods section.
- Methods, Questionnaire: What do the 12 pages refer to? It is an online-based survey. Do you mean 12 “screen pages”?
- Lines 148-150: The meaning of this sentence is not clear to me.
- Line 153: Have respondents been included who answered any question or all questions?
- Line 161: The fact that the majority of respondents were from maximum care hospitals is due to the recruitment process. This should be addressed in the limitations section.
- Lines 165-166: From my point of view, the time which was needed to spent for answering the questionnaire should be part of the methods section.
- Lines 193-202: A reference to Figure 1 is missing.
- Line 270: The abbreviation “NIV” has not been introduced before.
- Lines 310-311: This sentence is already an interpretation and not a summary as all the other sentences around this within the paragraph.
- Line 315: The citation is not correct.
-   Several references are incomplete, see e.g. 1, 7, 8, 27

In addition, the whole manuscript needs a thorough proof-reading. Furthermore, many parts of the manuscripts sound as if they were directly translated from German language to English. Maybe it is advisable to let a native speaker check the whole manuscript.

I have collected already several spelling mistakes here:

- Line 138: You can delete “a”.
- Lines 160 and 369: Please make the correction from a capital P in “participants” to a normal p.
- Line 177: Please correct “medical societies”.
- Line 185 and others: There are some mistakes in spelling the English name of the institute. It should read as “Robert Koch-Institute”.
- Line 215: Delete “s”.
- Line 216: There are two full stops at the end of the sentence.
- Line 432: One “with” should be deleted.
- Lines 443-444: There are inconsistencies in spelling “COVID-19”.
- Line 449: It should be “to” and not “do”.

Reviewer 2 Report

Thank you for the opportunity to review this interesting manuscript. In this study the authors aim to show the results of their survey to assess the use of different sources of information by different members of a German ICU regarding standards of care for COVID-19 patients. The authors show interesting differences between professions in relying on medical journals, journals of medical societies and everyday/social media. While the results are quite interesting and have the potential to inform how evolving standards of care are disseminated to different members of the medical team, the manuscript requires significant modifications.

- There extensive typographical errors throughout the manuscript that need to be revised.

- Both the results and discussion sections are long, broken down to multiple small paragraphs and difficult to follow. The results show many sections of all the study findings and many are repeated in the tables. The text of the results would be easier to follow if it were focused on the results of interest while pointing to the tables for the remainder. Similarly, the discussion is very difficult to follow.

- Both figures are lacking figure legends and only have figure titles. The text inside the figures is not readable.

- The tables require significant modifications of the wording and formatting to be readable

- IQR in the tables are reported as a single number. The IQR should show the range between the quartiles and so cannot be reported as a single value

- Several abbreviations throughout the manuscript as AWMF, MAGICapp, MD to name a few are not defined in the text

- While reporting the U, Z, p value and r are comprehensive, they are not necessary and make the results distracting and difficult to follow

Round 2

Reviewer 1 Report

All recommendations have been adressed.

Reviewer 2 Report

Thank you for the opportunity to review the revised version of the manuscript and the authors responses. In this version the authors have addressed the majority of the comments. The manuscript is now clearer and easier to follow.